# Consecutive treatments of methamphetamine promote the development of cardiac pathological symptoms in zebrafish

Jimmy Zhang[1], Anh H. Nguyen[2,3], Daniel Jilani[2], Ramses Seferino Trigo Torres[1], Lauren Schmiess-Heine[2], Tai Le[1], Xing Xia[2], Hung Cao[1,2,3]*

1 Department of Biomedical Engineering, University of California-Irvine, Irvine, CA, United States of America, 2 Department of Electrical Engineering and Computer Science, University of California-Irvine, Irvine, CA, United States of America, 3 Sensoriis, Inc., Edmonds, WA, United States of America

* hungcao@uci.edu

**Data Availability Statement:** All relevant data are within the paper and its Supporting Information

## Abstract

Chronic methamphetamine use, a widespread drug epidemic, has been associated with cardiac morphological and electrical remodeling, leading to the development of numerous cardiovascular diseases. While methamphetamine has been documented to induce arrhythmia, most results originate from clinical trials from users who experienced different durations of methamphetamine abuse, providing no documentation on the use of methamphetamine in standardized settings. Additionally, the underlying molecular mechanism on how methamphetamine affects the cardiovascular system remains elusive. A relationship was sought between cardiotoxicity and arrhythmia with associated methamphetamine abuse in zebrafish to identify and to understand the adverse cardiac symptoms associated with methamphetamine. Zebrafish were first treated with methamphetamine 3 times a week over a 2-week duration. Immediately after treatment, zebrafish underwent electrocardiogram (ECG) measurement using an in-house developed acquisition system for electrophysiological analysis. Subsequent analyses of cAMP expression and $Ca^{2+}$ regulation in zebrafish cardiomyocytes were conducted. cAMP is vital to development of myocardial fibrosis and arrhythmia, prominent symptoms in the development of cardiovascular diseases. $Ca^{2+}$ dysregulation is also a factor in inducing arrhythmias. During the first week of treatment, zebrafish that were administered with methamphetamine displayed a decrease in heart rate, which persisted throughout the second week and remained significantly lower than the heart rate of untreated fish. Results also indicate an increased heart rate variability during the early stage of treatment followed by a decrease in the late stage for methamphetamine-treated fish over the duration of the experiment, suggesting a biphasic response to methamphetamine exposure. Methamphetamine-treated fish also exhibited reduced QTc intervals throughout the experiment. Results from the cAMP and $Ca^{2+}$ assays demonstrate that cAMP was upregulated and $Ca^{2+}$ was dysregulated in response to methamphetamine treatment. Collagenic assays indicated significant fibrotic response to methamphetamine

files. Additionally, data relevant to this study are uploaded to Dryad at [https://doi.org/10.7280/D1269D].

**Funding:** The authors would like to acknowledge the financial support from the National Science Foundation (NSF, https://www.nsf.gov/) CAREER Award #1917105 (H.C.) and the NSF #1936519 (H. C). The funders had no role in study design, data collection and analysis, decision to publish, or preparation of the manuscript.

**Competing interests:** The authors have declared that no competing interests exist.

treatment. These results provide potential insight into the role of methamphetamine in the development of fibrosis and arrhythmia due to downstream effectors of cAMP.

## Introduction

Methamphetamines are sympathomimetic amines with a range of adverse effects upon multiple organ systems. Based around a phenylethylamine core, methamphetamine (Meth) and its analog, *d*-amphetamine, have high affinity with transporters associated with catecholamine signaling, significantly increasing the number of neurotransmitters such as dopamine and norepinephrine [1]. Unlike Meth, *d*-amphetamine has been prescribed as medication to treat neurological disorders such as attention deficit hyperactivity disorder (ADHD) and narcolepsy [2]. A possible reason for limited legal Meth use is the addition of the *N*-methyl group compared to amphetamine, which has been shown to confer better penetration through the blood-brain barrier for Meth, leading to stronger and more addictive responses [3]. Meth has been shown to induce heightened catecholamine response by promoting catecholamine release, preventing their reuptake, and destabilizing their levels [4, 5]. Thus, Meth is responsible for numerous neurotoxic symptoms, including potential neuronal apoptosis, decreased immune response, and associated memory deterioration [5–8]. Given that the elucidation of the direct mechanism of Meth was on neurological response, the major focus in researching treatments for Meth-related abuse has been associated with neurological modulation. Therefore, less attention was given to researching the direct mechanism of Meth in other physiological systems, such as the cardiovascular system.

Cardiotoxicity is one of the most adverse consequences of Meth abuse, leading to a notable increase of morbidity and mortality [1]. Cardiovascular complications are the second leading cause of death in Meth abusers. Cardiotoxicity can appear early in the course of the drug use and cause numerous significant effects, such as pulmonary hypertension, atherosclerosis, cardiac arrhythmias, acute coronary syndrome, and other associated cardiomyopathies [9]. Furthermore, a Meth 'binge' study in rats to determine long-term effects discovered that Meth decreased the sensitivity of nervous and cardiovascular physiology through successive treatments, implying the potential remodeling of electrophysiological responses through chronic Meth abuse [10]. Previous human case studies have determined that Meth abusers experienced increased ventricular tachycardia and QTc prolongation [11, 12]. However, case studies are generally retrospective, and there is a scarcity in animal studies regarding the effect of Meth on the actual initiation of arrhythmic symptoms [1]. Moreover, data on the underlying mechanism of cardiac dysfunction during drug abuse and the susceptibility of long-term cardiotoxic development are limited.

Despite the prevailing issue of Meth abuse, studies have shown that cardiac pathology induced by Meth can be attenuated and even reversed through the discontinuation of Meth use and the initiation of subsequent treatment [13]. A study in rats regarding the administration of Meth and eventual withdrawal revealed that the rats were able to recover from myocardial pathologic symptoms such as atrophy, fibrosis, and edema starting from 3 weeks after discontinued Meth administration [14]. A human case study indicated that attenuation of Meth use and subsequent therapy led to recovery from ventricular hypertrophy and ECG ST deviations [15]. While evidence of recovery from Meth abuse is promising for the development of future treatments, it is essential to conduct research to understand the specific mechanisms underlying Meth-induced cardiovascular pathologies. Therefore, a better understanding of the

cardiac dynamics of Meth abuse in zebrafish, a relevant model for human cardiac studies, may be vital for the future Meth-associated research.

In neurological studies associated with Meth, the drug-induced effect of G protein-coupled receptors (GPCRs) on subsequent neuropathology has been frequently investigated due to the receptors' association with neurotransmitters, hormones, and other neuromodulatory responses [16]. cAMP, a prominent secondary messenger within the GPCR signaling pathway, tends to be upregulated due to drug exposure, and it has been shown to influence sensitization and addiction to psychostimulant drugs [17]. However, given the ubiquity of GPCRs in numerous physiological systems, the effects of GPCRs on other symptoms of drug toxicity have also been explored. GPCRs have been demonstrated to be influential in the development of cardiovascular diseases. Studies have found that GPCRs induced increased $Ca^{2+}$ release and myocardial contractility, leading to a higher susceptibility to hypertrophy and cardiomyopathy [18]. GPCRs have also demonstrated the ability to modulate remodeling processes within the biological system, including epithelial-mesenchymal transition (EMT) and collagen deposition [19]. For example, TAAR1, a prominent GPCR targeted by Meth, was determined to be an integral player in the psychostimulant activity and addictive response of Meth [20]. However, TAAR1 has also been found outside of the neurological system, including the heart [21]. Therefore, cAMP and GPCRs may play a crucial role in the further elucidation of the effects of Meth on cardiac electrophysiology.

Although current research has not fully proven whether the single ventricular heart of zebrafish is comparable to the more complex ventricular conduction system found in higher vertebrates, zebrafish have been proposed as a versatile model system for researching biological applications due to similarities with humans pertaining to cardiac physiology. Zebrafish has also emerged as a high-throughput and low-cost animal research model that has been used for phenotype-driven drug screenings for new insights into chemical toxicity due to similar drug metabolism and genetic homology [22]. Our lab previously developed a method for measuring *in vivo* surface electrocardiography for adult zebrafish. This method was targeted for studies regarding irregular heartbeats and QT dysregulation, and the acquired results displayed remarkable electrophysiological similarities between zebrafish and humans [23]. As such, *in vivo* ECG for adult zebrafish would be a powerful tool for studies involving Meth-induced cardiac toxicity. In this study, we first demonstrate the potential of zebrafish ECG in the diagnosis of Meth-induced cardiotoxicity. We then sought to explain the results by conducting molecular analysis of the GPCR pathway within heart tissue, including the effect on fibrotic and $Ca^{2+}$ dysregulation, which attribute to cardiac toxicity. With the implementation of our ECG system in Meth studies, we hope to provide a new insight into the mechanisms of Meth-induced cardiotoxicity, further uncovering the multifactorial nature of Meth and assisting in the development of novel treatment methods.

## Materials and methods

### Zebrafish husbandry and preliminary surgical procedures

Wild-type zebrafish were housed in a custom built circulating fish rack system. The fish tanks were maintained at 28° C, ~pH 7.0, and these parameters were checked at least once daily. The system was equipped with four filters, a UV, carbon, and two particle filters. Zebrafish were kept under the 14:10 hour light/dark cycle. Zebrafish (AB-wild type strain) were approximately 6–12 months old at the onset of the experiment. Prior to Meth treatment, zebrafish underwent open chest surgery to improve subsequent ECG signal acquisition. Fish were first anesthetized with tricaine (200 mg/L) via immersion for approximately 5 minutes or until no opercular movement has been observed. Under a stereo microscope, scales (above the coelomic cavity

and the posterior site above the tail) were removed with forceps, exposing flesh, to allow more direct electrode contact. A small incision, ~2–3 mm, was made on the ventral surface of the fish above the heart. The incision cut through the chest wall and the heart was visible afterward. The fish were recovered in fresh fish water for a few minutes. The incision was not closed via suture, staple, clips, or glue as the chest wall and scales have been observed to regrow within 4 weeks. The zebrafish groups were then housed in separate tanks throughout the duration of the study. All zebrafish were checked every day for the first week and then at least 3 days/week in the second week until the experiment was concluded. Observations were made to see whether the fish displayed some abnormal activities (e.g., erratic swimming, strained breathing, bloating). The fish would be removed from the study if any abnormal activity was detected. In rare cases where extreme behavior/distress was seen, the fish would be euthanized. Euthanasia was conducted by immersing the fish in tricaine (250 mg/L) for at least 30 min after the last observed opercular movement. All fish carcasses will be frozen post-euthanasia. All zebrafish procedures were conducted in accordance to IACUC guidelines (#AUP-21-066 at University of California, Irvine).

## Drug preparation and methamphetamine treatment

The Meth solution for treatment (200 μM) was prepared by mixing the specified amount of Meth stock into regular fish water obtained from the fish rack system. Solutions were prepared fresh for each day of treatment. The Meth stock (1 mg/mL) was obtained from Sigma-Aldrich (MDL MFCD00056130). The solution (10 mL) was then placed in a small custom polydimethylsiloxane (PDMS) chamber suitable for housing one zebrafish. PDMS is a flexible and biocompatible polymer most commonly used in biosensors and implants [24]. Untreated fish were placed in the same custom chamber with regular fish water. Each zebrafish was treated in the designated treatment for 20 minutes before ECG recording, as determined in a previous study [25]. This treatment was conducted 3 times a week over a period of two weeks, following similar studies.

## ECG recording procedure and instrumental setup

ECG was obtained using the instrumental setup depicted in Fig 1A. Prior to placing the fish in the designated zebrafish station, the fish was first anesthetized in tricaine (200 mg/L) for approximately 5 minutes until the fish became unresponsive to external stimulus. Then, the fish was placed ventral side up in a precut crevice in the middle of the sponge. The sponge was then placed on a glass platform with the pin electrodes as positioned in Fig 1A, where the green working electrode was placed near the open incision on the chest while the yellow reference electrode was placed near the lower abdomen. Each fish underwent recording for approximately 1 minute before placing in regular fish water for recovery from anesthesia. Treatment concentrations and durations were approved by University of California-Irvine's IACUC.

The design of the recording system was modified from the initial design as described in the study by Yu *et al.* [26]. The pin electrodes were derived from isolating ends of jumper wires (Austor Part AMA-18-580) and stripping the outer insulator layer of the wire about 2 cm from the metal tip of the wires. The exposed copper wires underneath were then soldered to improve signal integrity and electrode longevity. The pin electrodes were embedded in a PDMS mold and were attached to alligator clips at the end of a cable wire leading to the differential AC amplifier (A-M Systems Model 1700). The AC amplifier bandpass filter settings were set to 1–1000 Hz. The signal from the zebrafish underwent 10000x amplification before undergoing digitalization with the data acquisition box (National Instruments Model USB-6001). The signal was then transmitted to a laptop (Dell Latitude E5470), where the processed

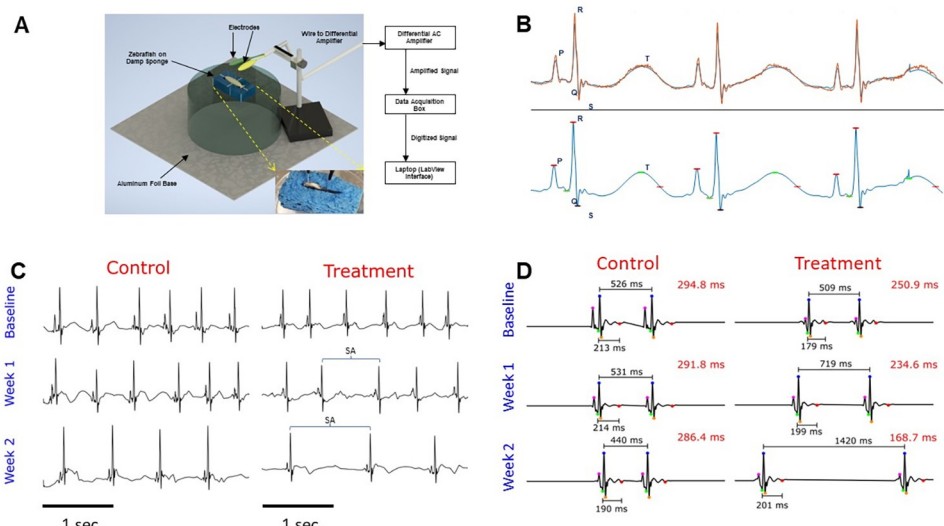

**Fig 1. Zebrafish ECG setup and representative ECG figures.** (A) The main figure depicts the general layout of zebrafish and electrodes during ECG recording. The working electrode is shown in green and contacts the chest cavity. The reference electrode is shown in yellow and contacts near the tail. This electrode setup connects with instruments outlined in the block diagram on the right, where signals are processed and displayed on the laptop. The inset provides a closer view of the positioning of electrodes on the zebrafish during recording. (B) Representations of ECG signal processing with labeled waveforms utilizing custom MATLAB software. The top displays the orange raw signal, while the bottom displays the blue processed signal. (C) These ECG signal figures were processed from both untreated (control, n = 6) and Meth-treated (treatment, n = 8) fish, taken during baseline, week 1, and week 2 of the study. The ECG waveforms (PQRST) were labelled on the first cycle of each figure. These figures depict that Meth treatment has significantly decreased heart rate over the duration of the study compared to no treatment. Additionally, Meth-treated zebrafish exhibited more pronounced bradyarrhythmia, as indicated by the blue brackets spanning across the occurrence of the bradyarrhythmia. Note that while control fish also exhibit bradyarrhythmia, the occurrences in Meth-treated fish were more pronounced. Figures shown represent 3 seconds of recording. Scale bar depicts 1 second. (D) These figures were produced by averaging all ECG segments from each recording, extrapolating the ECG waveforms in order to determine the T wave. The waveforms are depicted as follows: Magenta = P; Green = Q; Blue = R; Orange = S; and Red = end of T wave. The RR and QT intervals are labelled for all waveforms, and the calculated QTc interval is shown on the top right of each figure. In comparison between untreated (control, n = 6) and Meth-treated (treatment, n = 8) fish, treated fish exhibited progressively lower QTc throughout the duration of the experiment, likely due to decreasing heart rate (depicted as increasing RR interval).

ECG signal was displayed using the LabView interface. ECG signals were saved as text files using the LabView interface, where they can be accessed and visualized through MATLAB.

## ECG data collection and analysis

Before applying signal processing, the ECG signal was manually inspected and trimmed to remove segments with significant noise such as muscle twitching or electrical interference. A highpass and lowpass filter with 15 Hz and 70 Hz cut-off frequencies were applied to the ECG signal to remove respiratory, motion, and electrical artifacts. To accentuate the R wave morphology, a ricker wavelet with a central frequency of 25 Hz was convolved with the filtered signal. To better detect the R waves, the difference between adjacent samples was derived, squared, and smoothed using a 15-point moving average filter:

$$Y = \Phi\big((X_n - X_{n-1})^2\big) \tag{1}$$

where $X_n$ is the ECG indexed by $n$, $\phi$ is the moving average filter, and $Y$ is the resulting residual signal. A peak-finding algorithm was applied to the residual signal to determine the locations of the R waves in the ECG, and corrections were applied for mapping inaccuracies. The P, Q,

and S waves were detected by identifying the highest maxima and lowest minima preceding and following the R wave. Given the energy of the T wave was usually weak compared to the noise, the PQRST waveforms of the entire segment were averaged in order to extrapolate the general morphology of the PQRST, which produces an identifiable T wave. The T wave was identified as the highest point within a time range 125 to 175 milliseconds (ms) following the R wave, based on previous work [27]. These annotations were manually inspected and corrected after automatic detection. Annotated ECG signals with waveform detections are shown in Fig 1B. After annotating all of the waveforms in the signals, parameters including the heart rate (HR), heart rate variation (HRV), QRS interval, PR interval, and QTc interval were derived. The heart rate (in beats per minute) was calculated from the determined RR interval based on the following formula:

$$HR = \frac{60}{1000(RR)} \tag{2}$$

where RR is in ms.

The HRV, in ms, was determined by the root mean square of successive differences between normal heartbeats (RMSSD), given by the formula:

$$HRV = \sqrt{\frac{\sum_{i=1}^{N-1}\left(RR_{i+1} - RR_i\right)^2}{N-1}} \tag{3}$$

where $N$ represents the number of ECG cycles within each recording.

The corrected QT interval (QTc), in ms, was determined as shown in previous literature [28]:

$$QTc = \frac{QT}{\sqrt{RR}} \tag{4}$$

## Isolation of zebrafish cardiomyocytes

Zebrafish cardiomyocytes were isolated according to the protocol presented by Sander *et al* [29]. Briefly, 6 zebrafish hearts were first excised from anesthetized fish via incision with a pair of forceps. After incubating in heparin buffer immediately after excision, the hearts were placed in a digestion buffer (1x PBS, 10 mM HEPES, 30 mM taurine, 5.5 mM glucose, 10 mM BDM (butanedione monoxime, a contraction inhibitor), 12.5 μM CaCl$_2$, 5 mg/mL collagenases II and IV) for 2 hours in a thermomixer set at 32˚C and 800 rpm. The digested tissue was then washed with a sequential series of stopping buffers (1x PBS, 10 mM HEPES, 30 mM taurine, 5.5 mM glucose, 10 mM BDM, 5–10% FBS, 12.5–1000 μM CaCl$_2$). After washing, the isolated cardiomyocytes were plated on 2 wells in a 96-well plate. The plating medium consisted of DMEM with 2 mM glutamine, 5 mM BDM, 5% FBS, 100 U/mL penicillin-streptomycin, and 1:500 Normocin (InvivoGen).

## Cloning, cell culture and transfection

Rat *TAAR1* gene was amplified from total cDNA using the set of primers: (BamHI) ACCATG**G**CATCTTTGCCACAATAGCGC and (NotI) ACAAAAATAACTTAGACCTAGATGAATCT. After amplification, the *TAAR1* gene was cloned into pcDNA3.1 Zeo (+) (Invitrogen) and transformed in E. coli DH5α. The cloned plasmid product was sequenced (Sanger method) using the Sanger Sequencing Kit (Applied Biosystems). For transient expression, the plasmid (10 μg) was transfected to human embryonic kidney cells (HEK293) transiently expressing the recombinant TAAR1 protein. The HEK293 cells

were maintained in DMEM containing 10% FBS and 1% penicillin/streptomycin at 37°C in a 5% $CO_2$ incubator. After 24 h, the cells were maintained in the media containing zeocin (100 μg/mL) for a stable expression of *TAAR1* in HEK293 cells. The transfected HEK293 cells were used in the subsequent cAMP assay and calcium assay.

## GloSensor cAMP assay

The HEK293 and isolated zebrafish cardiomyocytes were cultured in DMEM (10% FBS and 1% PS) in a poly-D-lysine pre-coated 96 well microplate and incubated in the $CO_2$ incubator at 37°C. 0.2 mL of cardiomyocyte selective growth supplement (Sciencell) was added to the cardiomyocyte culture. The pGloSensor-22F-cAMP plasmid (10 μg was transfected into HEK293 cells and isolated zebrafish cardiomyocytes ($1.5 \times 10^4$ cells) using the Lipofectamine 3000 reagent as the manufacturer described. 48 hours after transfection, cells expressing the plasmid (15,000 cells/well) were collected. The desiderated cell number was incubated in equilibration medium containing a 2% (v/v) GloSensor cAMP reagent (Luciferin) stock solution, 10% FBS and 88% $CO_2$-independent medium in 2 hours at 37°C according to the manufacturer's instructions. The cells were dispensed in wells of 96-well plate and a basal signal was obtained before treating with Meth with doses from $10^{-10}$ to $10^{-2}$ M was added at room temperature (~25°C) before luminescence detection. The original cell population isolated from the zebrafish heart was separated into the respective groups for treatment with different doses. EPPTB (0.1 μM) was also utilized as a selective TAAR1 and GPCR antagonist in this assay [30]. Therefore, according to the proposed pathway seen in Fig 3A, EPPTB would inhibit cAMP expression.

## FLIPR calcium assay

Two days after transfection, cells were washed with FLIPR buffer (1x HBSS, 20 mM HEPES, 2.5 mM probenecid, pH 7.4), loaded with the calcium-sensitive fluorophore Fluo-3 (Thermo-Fisher) for 1.5 h at 37°C, 5% $CO_2$. During the incubation, two separate 96-well polypropylene compound plates were prepared. Meth was prepared with 10-point concentration-response curve dissolved in the buffer. EPPTB was prepared in 1% DMSO in FLIPR buffer. After the incubation with Fluo-3 dyes on the assay plate, different Meth doses were added to the assay plate and incubated for 15 min at 37°C. Subsequently, the assay plate was read with a Fluorescent Imaging Plate Reader (FLIPR) Tetra (Molecular Devices). Data of calcium-responsive changes in fluorescence were collected every second over a 60-second time period. Regarding the antagonist assay, the assay plate with EPPTB were incubated for 15 min before monitoring fluorescence.

## Collagen assay

The level of collagen in tissues was measured by the collagen assay kit (Sigma Aldrich) following standard protocol. Briefly, collagen in samples was first enzymatically digested into collagen peptides in master reaction mix including 35 μL buffer and 0.5 μl Collagen I. The reaction mix was incubated at 37°C for 60 min. Subsequently, 40 μL of Dye Reagent to wells and incubated at 37°C for 10 min. The collagen levels are determined by reading fluorescence at 465 nm.

## Histochemical staining and immunofluorescent staining for collagen

After treatment of Meth, zebrafish hearts were isolated as described in previous literature [29]. Briefly, the fish were anesthetized by tricaine before undergoing chest incisions. The heart was

then located and excised, which were then placed in a solution of perfusion buffer (10 mM HEPES, 30 mM taurine, 5.5 mM glucose and 10 mM BDM in 1x PBS solution). After excision of all hearts, they were subsequently fixed by 4% formaldehyde and cryo-sectioned with a cryostat. The tissue slices were placed onto frosted microscope slides (Thermofisher Scientific) and underwent Masson's Trichrome staining (following the provided protocol from American Master Tech, TKMTR2). Myocardial tissue is stained red, and collagenous tissue is stained blue. For immunofluorescence, mouse anti-collagen I antibody (Novus Biologicals) and rabbit anti-collagen III antibody were incubated overnight on cryosections of cardiac tissues, followed by staining for 2 h with goat-anti mouse IgG conjugated Alexa 568 (Abcam) and donkey-anti rabbit IgG conjugated FITC (Thermo Fisher). Sections were mounted in Antifade Mounting Medium with 4′,6-diamidino-2-phenylindole (DAPI) (Thermo Fisher) and mounted in Antifade Mounting Medium. Fluorescent imaging was taken with the Keyence Digital Microscope (BZ-X800) system.

## Statistical analysis

Statistical analysis was conducted via the JMP Suite, a statistical data analysis tool derived from SAS. All parameters derived from data analysis (*i.e.*, HR, HRV, QRS, PR, QTc) were averaged within each experimental group. Statistical significance was determined by the one-way ANOVA test between experimental groups with significance level $p<0.05$. Data in figures were plotted as mean ± standard error (SE).

Outputs of the cAMP and calcium assays were expressed as relative luminescence units (RLU) and relative fluorescent units (RFU), respectively. A non-linear regression (Prism 8.0 GraphPAD Software, San Diego, CA, USA) was used to quantify methamphetamine potency for EC50 value calculations. EC50 is the concentration of agonists required to produce 50% of the maximum effect. The maximum effect obtained in the cAMP and calcium assays were approximately 1400000 RLU and 12000 RFU, respectively. Each concentration was tested three times in triplicate, and the values were given as mean ± SE. Significance was determined via the Student's T-test.

## Results

### Methamphetamine induced significant ECG changes over the course of 2 weeks

Using the ECG setup designed in our lab (Fig 1A), we acquired ECG signals over a two-week period. The control (n = 6) and Meth-treated (n = 8) groups were immersed in the designated solutions for 20 minutes for 3 instances per week, and their ECG were subsequently acquired after treatment. The raw signals were then processed to eliminate external noise, and the PQRST waveforms were labelled on the processed ECG signals. Representative raw signals (orange) and annotated processed signals (blue) are displayed in Fig 1B. Relevant ECG parameters including HR, HRV, QRS, QTc, and PR intervals were then quantified. The day number was defined as follows: day 1 corresponded to the first day of Meth treatment, and subsequent days were numbered accordingly. Fig 1C and 1D comprise ECG diagrams obtained during baseline, week 1, and week 2 of the treatment regimen. Fig 1C represents individual ECG signals after filtering and smoothing. Fig 1D represents the averaged outcome of all ECG signals for the specific experimental group and time period, developed for the detection of the T wave. The representative ECG figures indicate a progressive decrease in heart rate for the Meth-treated group with instances of sinus arrest highlighted by blue brackets (Fig 1C). The QTc interval also displayed a decrease in the treated group (Fig 1D). Calculated ECG parameters

are displayed in Fig 2. All ECG parameters were verified to display no significant differences during baseline measurement between the experimental groups. Two trials were conducted, and the results of the second trial are presented in S1 Fig. Additionally, the results also indicate that there were no significant differences in QRS duration between the Meth-treated fish and the untreated (control) fish throughout the duration of the study (Fig 2E).

The heart rate (HR) for treated fish decreased throughout the course of the treatment, exhibiting signs of bradyarrhythmia (Fig 1C). In the first trial, the HR presented significant differences at the end of the first week of the treatment (Fig 2A). The HRs on day 5 (p = 0.008) were 79.77 BPM (95% CI, 65.43–94.11) and 111.9 BPM (95% CI, 95.31–128.4) for treated and untreated fish, respectively. The HR stabilized in the second week (days 10 (p = 0.004) and 12 (p = 0.012)), displaying a consistent decrease through the remainder of the study. The HRs for treated fish were 71.17 BPM (95% CI, 56.71–85.63) for day 10 and 68.99 BPM (95% CI, 55.55–82.44) for day 12. These were in contrast with the HRs seen in untreated fish, which displayed 107.9 BPM (95% CI, 91.21–124.6) and 95.89 BPM (95% CI, 81.37–110.4) for days 10–12, respectively. In the second trial, the HR for treated fish displayed a significant decrease on the third day of treatment and remained lower than those of untreated fish throughout the course of the experiment (S1A Fig).

Throughout the course of the study, the heart rate variation (HRV) displayed a biphasic trend for the treated fish (Fig 2B). The HRV between Meth-treated fish and control fish exhibited no differences on days 1 and 3, but Meth seemed to have induced an increase in HRV during the first week of treatment. Treated fish exhibited a significant increase in HRV during day 5 (p = 0.004), reaching up to 338.6 ms (95% CI, 251.5–425.6) vs 121.7 ms (95% CI, 21.20–222.3) seen in untreated fish. This also corresponds to the maximum HRV attained by the treated fish during the study. The increase in HRV may be attributed to the presence of sinus arrest as indicated by the blue brackets in Fig 1C. The HRV decreased during the second week of treatment, although it still remained above the HRV for untreated fish. On day 12 (p = 0.851), the heart rate variation for Meth-treated fish was 181.2 ms (95% CI, 48.12–314.3)

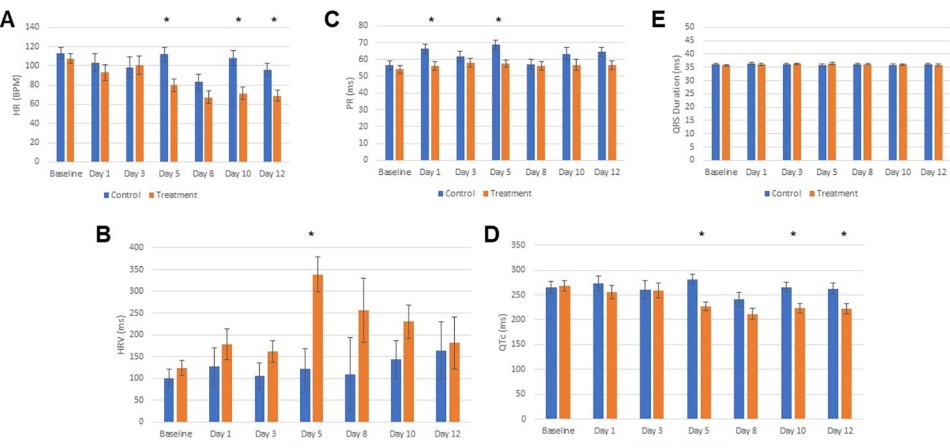

**Fig 2. Meth induces a biphasic and depressive effect on cardiac electrophysiology.** Tabulated averages from all ECG data acquired from both untreated (control, n = 6) and Meth-treated (treatment, n = 8) groups across the 2-week study. (A) Meth-treated fish displayed decreased heart rate compared to the untreated fish starting from the end of the week 1. (B) Meth-treated fish exhibited a biphasic trend in heart rate variation (HRV) throughout the duration of treatment, reaching a peak at the end of week 1 before decreasing during week 2. (C) Meth treatment induced a significant decrease in PR interval during week 1 but not week 2 of treatment. (D) Meth treatment induced a significant decrease in QTc during the end of week 1 before maintaining the depressed QTc throughout week 2. (E) Meth treatment did not exhibit a change in QRS duration. * denotes p<0.05.

compared to 164.1 ms (95% CI, 20.32–307.8) seen in untreated fish. Results from the second trial displayed a similar biphasic trend for HRV within treated fish (S1B Fig). HRV exhibited a steady increase until Day 8 of treatment before subsequently decreasing through the end of the study.

In the first trial, Meth induced significant changes in the PR interval during the first week of treatment (Fig 2C). On day 1 (p = 0.019), the Meth-treated fish exhibited a PR interval of 56.00 ms (95% CI, 50.55–61.45), compared to the PR interval of untreated fish, which was 66.37 ms (95% CI, 60.08–72.66). For day 5 (p = 0.007), the PR intervals for Meth-treated fish and untreated fish were 57.55 ms (95% CI, 52.50–62.59) and 68.91 ms (95% CI, 63.09–74.74), respectively. There were no significant differences found during the second week of treatment. Interestingly, the differences in PR interval were not significantly evident during the second trial (S1C Fig).

The QTc interval displayed a significant decrease during the first week for Meth-treated fish in both trials(Fig 2D and S1D Fig). The most significant decrease occurred on day 5 (p = 0.001), with QTc interval being 226.8 ms (95% CI, 208.4–245.3) and 281.5 ms (95% CI, 260.2–302.8) for treated and untreated fish, respectively. The QTc interval stabilized during the second week, generally maintaining the significant decrease for Meth-treated fish. On day 12 (p = 0.017), the QTc interval was 222.2 ms (95% CI, 200.6–243.8) and 262.7 ms (95% CI, 239.4–286.1) for treated and untreated fish, respectively.

## Methamphetamine treatment leads to increased expression of cAMP and $Ca^{2+}$ in a TAAR1-mediated, dose-dependent manner

To help uncover the direct molecular mechanism in the induction of Meth-induced cardiotoxicity, zebrafish cardiomyocytes (n = 6 zebrafish) were isolated and treated with Meth to determine Meth induced cAMP expression via the GloSensor cAMP assay, which is involved in regulating GPCR pathways. TAAR1-overexpressed HEK293 cells have also been produced to complement the assay as the positive control, and original HEK293 cells served as the negative control. Fig 3A depicts the proposed GPCR/cAMP pathway that enables Meth-induced cardiotoxicity. In previous studies, Meth has been discovered to bind to GPCRs such as TAAR1 to trigger the upregulation of cAMP in cardiomyocytes [31]. cAMP downstream signaling may be linked with the onset of cardiac pathology, such as fibrotic dysregulation via lysyl oxidase and arrhythmia via CaMKII. The results from the cAMP assay indicate that Meth led to a dose-dependent upregulation in cAMP expression within zebrafish cardiomyocytes (Fig 3B). Additionally, TAAR1-overexpressed HEK293 cells displayed a greater increase in cAMP expression than original HEK293 cells, indicating that TAAR1 mediated cAMP expression due to Meth exposure. In a similar fashion, $Ca^{2+}$ was also upregulated due to Meth exposure within zebrafish cardiomyocytes (Fig 3C). TAAR1-overexpressed HEK293 cells displayed a greater increase in $Ca^{2+}$ concentration than original HEK293 cells, indicating that TAAR1 also mediated $Ca^{2+}$ dysregulation due to Meth exposure. These results are further corroborated by experiments involving EPPTB, a TAAR1 antagonist. As indicated in Fig 3D, 3E, EPPTB attenuated Meth-induced cAMP and $Ca^{2+}$ upregulation in a dose-dependent manner. This also suggests that Meth-induced cardiotoxicity involving cAMP and $Ca^{2+}$ dysregulation may be attenuated by targeting TAAR1 and the GPCR pathway.

## Methamphetamine treatment produce excessive fibrosis in zebrafish cardiac tissue

To determine if Meth induces fibrotic dysregulation, which was delineated above as one of the potential cardiotoxic factors contributing to arrhythmias, Masson's Trichrome staining for collagen and collagen type I immunological staining were conducted on cardiac tissue

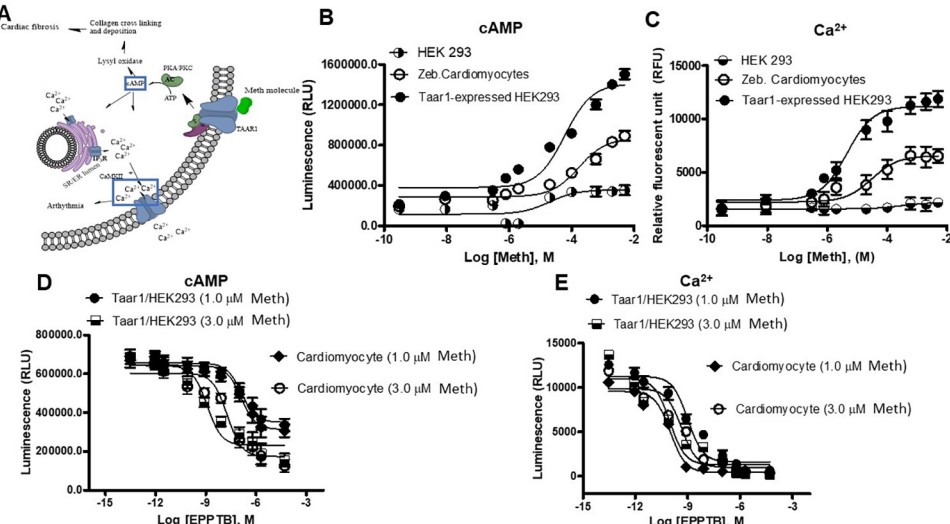

**Fig 3. Zebrafish cardiomyocytes exhibit a TAAR1-mediated increase in cAMP expression and $Ca^{2+}$ concentration due to meth treatment.** (A) Proposed mechanism of the pathologic effects of Meth on cardiomyocytes, including the upregulation of cardiac fibrosis via lysyl oxidase and the increased frequency of arrhythmia via the calmodulin CaMKII. cAMP, the upstream factor for both processes, and $Ca^{2+}$, the ion modulated by CaMKII, are both highlighted to indicate that they were investigated in this study. (B) Detection of cAMP expression after Meth treatment from zebrafish cardiomyocytes, HEK293, and TAAR1-overexpressed HEK293 cells via the GloSensor cAMP assay. TAAR1-overexpressed HEK293 cells served as the positive control for this assay. HEK293 cells served as the negative control. Results display that Meth induced dose-dependent cAMP expression in zebrafish cardiomyocytes. cAMP expression from TAAR1-overexpressed HEK293 cells exhibited a greater dose-dependent increase, demonstrating TAAR1-mediated cAMP expression due to Meth treatment. (C) Detection of $Ca^{2+}$ after Meth treatment from zebrafish cardiomyocytes, HEK293, and TAAR1-overexpressed HEK293 cells. Results display that Meth increased $Ca^{2+}$ concentration within zebrafish cardiomyocytes in a dose-dependent manner. $Ca^{2+}$ expression from TAAR1-overexpressed HEK293 cells exhibited a greater dose-dependent increase, demonstrating that TAAR1 mediates $Ca^{2+}$ concentration due to Meth treatment. (D-E) Results from the dose-response experiment with EPPTB, an inhibitor of TAAR1, in the presence of Meth, further corroborating previous results of TAAR1-mediated increases of cAMP expression (D) and $Ca^{2+}$ concentration (E).

obtained from both untreated and Meth-treated fish (n = 6 per group). Images obtained from Masson's Trichrome staining (Fig 4A, 4B) revealed a higher presence of collagen deposits in Meth-treated cardiac tissue, as highlighted in the dotted boxes. Immunological staining (Fig 4C, 4D) further revealed a higher presence of collagen type I with Meth-treated tissue. Collagen fluorescent assay of purified protein samples revealed that the collagen content increases from Meth treatment in a dose-dependent manner, displaying the highest difference at the highest concentration of Meth (Fig 4E). An expression profile of genes associated with fibrosis was also conducted, and the results determined that Meth-treated tissue displayed significantly higher expression (p<0.005) in lysyl oxidase (*LOX*) and lysyl hydroxylase (*PLOD*) (Fig 4F). This result signifies that the lysyl oxidase family of proteins was upregulated in response to Meth treatment. While the other genes involved in the profibrotic response (*COL1A1*, *COL3A1*, *MMP1*, and *TMP1*) did not display significance, their expressions have also shown marginal increases in Meth-treated cardiac tissue over its untreated counterpart. Overall, these results display an uptick in the fibrotic response due to Meth treatment, further outlining Meth-associated cardiotoxicity in relation to the GPCR/cAMP pathway.

## Discussion

The methamphetamine epidemic continues to fester worldwide, and cardiovascular diseases remain leading causes of death for methamphetamine abusers. Utilizing animal models to

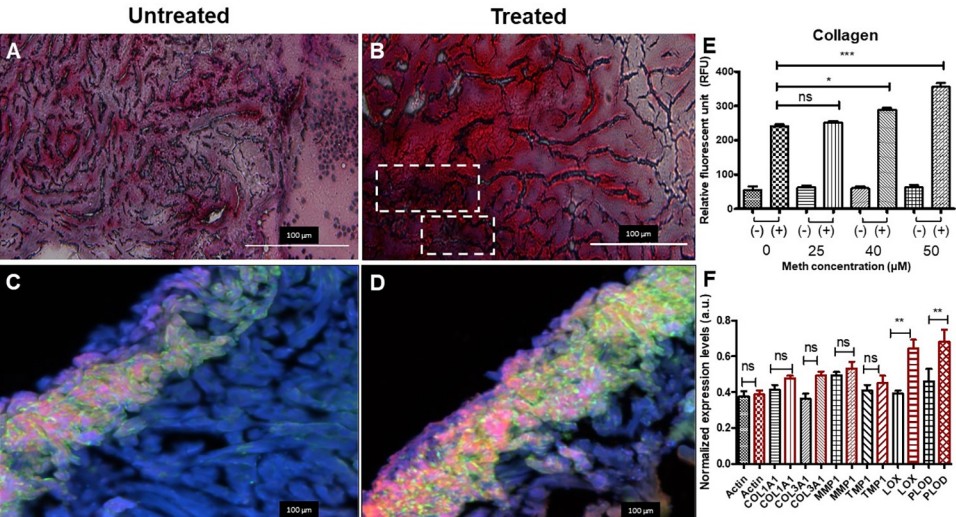

**Fig 4. Meth results in increased fibrotic response in zebrafish cardiac tissue.** (A-B) Masson's Trichrome Staining of cardiac tissue obtained from untreated zebrafish (n = 6) and Meth-treated zebrafish (n = 6). Myocardial tissue is stained in red, and collagen is stained in blue. Areas of collagen deposits are labeled by white dashed boxes in Meth-treated cardiac tissue. (C-D) Collagen type I immunological staining of untreated and treated cardiac tissue. Presence of lighter colors (*i.e.*, pink, yellow, green) indicates the presence of collagen. Meth-treated tissue displays higher amounts of collagen I. (E) Collagen fluorescent assay conducted on protein samples treated at various concentrations of Meth. Positive symbol (+) signifies collagenase I treatment and negative symbol signifies no treatment. Results indicate that collagen content increases due to Meth treatment in a dose-dependent manner. (F) Expression analysis of genes associated with fibrosis, including the family of lysyl oxidases. (*COL1A1* = collagen I; *COL3A1* = collagen III; *MMP1* = matrix metalloproteinase I; *TMP1* = thymidylate kinase; *LOX* = lysyl oxidase; *PLOD* = lysyl hydroxylase) Meth-treated tissue exhibited significantly higher expression of lysyl oxidase and lysyl hydroxylase, associated with the GPCR pathway. Expression of other genes displayed marginal increases in treated tissue, indicating higher instance of fibrosis. * denotes p<0.05. ** denotes p<0.005. *** denotes p<0.0005.

study cardiovascular associated mechanisms could be critical in devising treatments for Meth associated diseases. The zebrafish is an excellent model for drug screening studies due to high fecundity, low maintenance, and similar genetic homology to that in humans. As the zebrafish model is constantly evolving, studies have continued to delineate the applicability of the zebrafish in human medical research. During the initial conception of this study, we sought to 1. Establish the zebrafish model as an adequate model of drug screening for cardiotoxic effects, and 2. Characterize the electrophysiological abnormalities due to meth administration in a controlled environment. Utilizing our custom-designed zebrafish ECG acquisition system in our lab, we were able to acquire ECG from zebrafish during the two-week treatment period with Meth. Based on our results, we determined several significant ECG changes occurring between the Meth-treated and untreated fish. The progressive decrease in heart rate for treated zebrafish, while contradictory to the results seen in human clinical studies, was actually consistent with the results seen in previous animal model studies, including those performed on rats, monkeys, and zebrafish embryos [10, 32, 33]. Those previous studies suggested that the decrease in heart rate due to Meth administration may be attributed to the baroreceptor reflex, a homeostatic response to the increase in blood pressure [34]. Schindler *et al.* documented a consistent blood pressure increase with the dosage of Meth administered in squirrel monkeys [32]. However, they noted that heart rate modulation from Meth administration may be biphasic, as lower Meth concentrations induces tachycardia, while higher Meth concentrations induces bradycardia. This suggests that there is a critical Meth concentration where it achieves a maximum catecholaminergic effect without triggering a significant baroreceptor reflex. The

decrease in PR interval due to Meth treatment suggests quicker atrial conduction, which is consistent with the mechanism of Meth to induce ventricular tachycardia [35]. The heart rate variability exhibited a peak at the end of the first week before decreasing during the second week.

HRV is used in cardiac physiology as a measure of healthy function and as a potential parameter for determining cardiovascular diseases such as sudden cardiac death [36]. More importantly, HRV may also provide insight into the brain-heart axis and how the autonomic nervous system impacts cardiac function, critical to the analysis of stimulant drugs such as Meth [37]. The initial increase in HRV may likely be due to the presence of sinus arrest in Meth-treated fish. As shown in Fig 1C, ECG acquired from Meth-treated fish displayed episodes of sinus arrest, where an instance of cardiac conduction normally present in regular sinus rhythm is absent. The HRV increase may also be attributed to the baroreceptor reflex, as the reflex would naturally adjust the heart rate in order to maintain stability in blood pressure and cardiac output. Most notably, the decrease in HRV during the second week of treatment is most likely associated with cardiac tissue damage and inflammation, consistent with the symptoms seen in Meth-induced cardiomyopathy [38, 39]. This is corroborated with research indicating that persistent hypertension induced a decrease in HRV, presenting the long-term pathophysiological effects of Meth [40]. Additionally, previous case studies involving Meth and amphetamine abusers indicated that the presence of elevated cardiac biomarkers such as troponin I and creatine kinase-MB, further suggesting that common symptoms of Meth use may be associated with cardiac damage [41–43]. Overall, the HRV results from this study suggest the biphasic nature behind the mechanism of Meth, where the cardiovascular system may initially respond to the effects of the drug before sustaining damage after a period of persistent exposure.

Additionally, the QTc interval, associated with ventricular contraction and proper heart function, decreased for Meth-treated fish, displaying the most pronounced differences in the second week of treatment. Initial observation suggested that the decrease in QTc interval seemed to be associated with the decrease in heart rate (or increase in RR interval, as depicted in Fig 1D. The QT/HR relation has been widely documented, leading to the creation of the corrected QT interval to account for the effect of heart rate changes on the QT interval [44]. Nevertheless, the results obtained from this study remained contrary to the results seen in previous case studies, where Meth abusers tended to exhibit prolonged QTc intervals in response to Meth intake [12, 45]. However, these case studies utilized data from humans who were already predisposed to Meth for varying periods of time, usually to the point of drug dependence. Therefore, QTc prolongation could potentially be a symptom seen in the later stages of Meth-induced cardiotoxicity. Previous clinical trial research has associated the development of QTc prolongation with ventricular tachycardia and cardiomyopathy, which are common symptoms of Meth abuse [38, 46, 47]. QTc shortening seen in this study may also suggest that calcium channels are downregulated by Meth, as decreased calcium influx also reduces action potential duration. This is corroborated by ion channel expression analysis from rats suggesting that Meth reduces calcium channel expression [48]. Potassium channel expression may also play a role in analyzing the effect of Meth on modulating the QT intervals. Numerous anticonvulsants and antiarrhythmics have been known to shorten QTc by upregulating potassium channel function [49]. Moreover, Meth has been shown to induce upregulation of potassium channels in the brain in relation to its neuropathic effects, suggesting that Meth has the potential to modulate potassium channels in other pathologies [50]. However, more research will be needed to elucidate the direct effects of Meth on ion channels *in vivo*, as patch clamp results from rat cardiomyocytes regarding the effect of Meth on calcium channels remain controversial [51, 52]. In general, ion channel analysis in cardiovascular pathology remains scarce.

We surmise that Meth might pose an antagonizing interaction with tricaine, the anesthetic agent used to acquire ECG. Tricaine usage was also mandatory for this study due to regulatory purpose. However, both Meth and tricaine have opposing mechanisms, as Meth is a stimulant while tricaine is an established anesthetic, known for preventing action potential firing by blocking voltage-gated sodium channels [53]. The data revealed that Meth-treated zebrafish exhibited significant decreases in PR interval on certain days in the early stage of treatment due to the excitatory properties of Meth. However, a closer inspection of the data indicated that the increase in PR interval for untreated fish was responsible for the significant change instead. Indeed, tricaine may induce a decrease in myocardial contractility, which results in a decrease in heart rate and increase in PR interval during zebrafish sedation [54, 55]. Additionally, zebrafish subjected to repeated tricaine treatment exhibited increased susceptibility to anesthetic effects [56]. Results from the first trial revealed a decrease in heart rate for untreated fish during the treatment period (day 8). However, the heart rate remained stable during the second trial, suggesting that the susceptibility to anesthetic effects may be varied between individual organisms. Meth may also induce arrhythmic instances, which could confound HRV measurements [57]. In the first trial of this study, insignificant but noticeable increases in HRV were evident in days 10 and 12 for untreated fish. However, this was not seen in the second trial. Meth studies involving other animal models have also suggested the confounding effect of anesthetics. Research concerning the hemodynamic response to Meth reported differing results, as a study conducted on anesthetized cats documented a decrease in blood pressure due to Meth administration, while a study conducted on conscious monkeys indicated an increase in blood pressure [32, 58]. It is unclear if the difference in results was due to the presence of anesthesia or an underlying combinatorial effect of Meth and the anesthetic agent. Research has also suggested that Meth confers a depressor effect in addition to the commonly known pressor effect, and the depressor effect may dominate for animals under anesthesia. Vaupel *et al.* reported a significant decrease in blood pressure after the onset of Meth administration in anesthetized rhesus monkeys, suggesting the presence of the depressor effect [59]. In addition, tricaine has been shown to induce augmented effects with other agents. Muntean *et al.* treated zebrafish larva with dopamine and verapamil under both tricaine-anesthetized and methylcellulose-embedded conditions, and their results indicate that the effects of the agents on myocardial calcium signaling and heart rate were greater in anesthetized fish than embedded fish [60]. This suggests that tricaine has the potential to induce drug-drug interactions with other agents to influence underlying electrophysiology. Therefore, future improvements should be implemented to reduce the effect of tricaine for zebrafish cardiotoxic studies, as tricaine could introduce confounding circumstances, especially when testing psychostimulants on zebrafish. Future studies should also seek to explain the effect of these chemical entities on ion channel function through analysis of sodium and calcium transients for zebrafish. These future experiments would determine the mechanism of methamphetamine in inducing cardiotoxicity as well as bolster the use of zebrafish as a suitable model for cardiotoxic studies.

GPCRs and one of their prominent secondary messengers, cAMP, have been attributed to modulate numerous neurological dysregulations due to Meth exposure [17, 61]. For example, studies have discovered that TAAR1 was significantly involved in the modulation of the physiological and addictive response to Meth [20]. As a result, subsequent research has targeted TAAR1 for developing genetic and pharmacological treatments for the methamphetamine epidemic [21, 22]. Due to ubiquitous nature of TAAR and GPCRs, this signaling pathway is also located in other physiological systems. Fehler *et al.* determined the presence of TAAR receptors within the aorta and their role in drug-associated vasoconstriction, which leads to cardiovascular conditions such as high blood pressure [62]. Interestingly, this process does not appear to be mediated by the neuronal system [63]. Recent research has indicated that GPCRs

may also play a role in producing detrimental cardiac effects of Meth, including arrhythmia, fibrosis, cardiomyopathy, and tissue remodeling [64, 65]. As depicted in Fig 3A, cAMP interfaces with numerous factors within the GPCR pathway that may induce such cardiovascular effects [66]. cAMP upregulation is known to lead to fibrosis via increased lysyl oxidase production, as lysyl oxidase plays a major role in collagen and ECM crosslinking [67]. cAMP is also involved in the modulation of CaMKII, principal in the maintenance of myocardial calcium ion homeostasis [68]. CaMKII dysregulation has been attributed to the development of cardiac pathologies, such as the regulation of cardiac extraction-contraction coupling and the activation of inflammatory and hypertrophic pathways [69]. Using the GloSensor cAMP assay, we demonstrated that the upregulation of cAMP occurred in a dose-dependent manner within zebrafish cardiomyocytes. This upregulation was inhibited by EPPTB, an antagonist of TAAR1 and cAMP. We also demonstrated that $Ca^{2+}$ was upregulated by Meth in a TAAR1-dependent fashion. Collagenic assays on untreated and Meth-treated zebrafish further indicated that Meth induced an increased fibrotic response in cardiac tissue, consistent with the concept that Meth-associated fibrosis led to the dysregulation in cardiac electrophysiology. The significant upregulation in the family of lysyl oxidase proteins further suggests that TAAR1 and GPCRs modulate this response. Overall, the results from this study indicate that Meth upregulated cAMP in zebrafish cardiomyocytes, causing dysregulation in $Ca^{2+}$ homeostasis and fibrotic response, suggesting that cAMP and GPCRs play a role in Meth-induced cardiotoxicity.

Additional research should be conducted to further understand the role of heart-brain axis due to Meth exposure, such as the link between neurotransmitter response and cAMP/GPCR expression to cardiovascular abnormalities, as well as an investigation on ion channel function in the heart after Meth administration The zebrafish model has already been utilized in numerous Meth studies, mostly related to behavioral studies due to the ability of Meth to disrupt dopamine release and reuptake, thus increasing dopamine expression [70]. Therefore, it would be intriguing to understand the role of dopamine in Meth-induced cardiotoxicity, as it would explain whether Meth-induced cardiotoxicity is caused by dopamine or through a direct effect from Meth. One consequence of dopamine response is the change in ion channel expression. For example, studies have shown the modulation of L-type calcium channels by Meth, but it is not fully understood whether Meth alters calcium channel function directly or via dopamine [51, 71, 72]. As mentioned earlier, ion channel modulation may also be integral in understanding the cardiac electrical remodeling induced by Meth. Meth has been determined to alter the expression and functionality of potassium and calcium within cardiomyocytes, which were correlated to Meth-associated arrhythmic events [51, 52]. These effects were not attributed to neuronal functions, indicating that separate mechanisms may also be in play for Meth-associated alterations in the neuronal and cardiovascular system.

The goal of this study is to provide elucidation into the effect of Meth on cardiac physiology and electrophysiology in a standardized, controlled setting. The generated results suggest that Meth induces a predominant depressor effect on cardiac electrophysiology most likely due to the baroreceptor reflex and cardiac damage. This was manifested as a progressive decrease in heart rate and eventual decrease in HRV. This effect persisted through the end of the two-week treatment, which may be a sign of cardiac damage as seen in Meth-induced cardiotoxicity. Molecular analysis suggested that the Meth exposure via cAMP upregulation leads to the development of fibrosis and arrhythmia.

## Supporting information

**S1 Fig. Second trial of electrophysiological analysis of meth treatment yields similar results.** A repeated trial of both control and treatment fish was conducted, and averages are

tabulated in the same format as shown in Fig 2. (A) Treatment fish displayed significantly decreased heart rate compared to control starting from day 3 of treatment (week 1). (B) Treatment fish exhibited a biphasic trend in heart rate variation (HRV), peaking at day 8 of treatment, corresponding to the beginning of week 2. (C) Treatment fish did not exhibit a significant difference in PR interval. (D) Treatment fish displayed a significant decrease in QTc during the end of week 1 and throughout week 2. (E) Treatment fish did not display a significant difference in QRS duration. * denotes $p < 0.05$.
(DOCX)

**S1 Data.**
(XLSX)

## Author Contributions

**Conceptualization:** Jimmy Zhang, Hung Cao.

**Data curation:** Jimmy Zhang, Daniel Jilani, Lauren Schmiess-Heine.

**Formal analysis:** Jimmy Zhang, Anh H. Nguyen.

**Funding acquisition:** Hung Cao.

**Investigation:** Jimmy Zhang, Daniel Jilani, Ramses Seferino Trigo Torres, Lauren Schmiess-Heine.

**Methodology:** Jimmy Zhang, Anh H. Nguyen, Daniel Jilani, Ramses Seferino Trigo Torres, Lauren Schmiess-Heine, Tai Le, Xing Xia.

**Project administration:** Jimmy Zhang, Hung Cao.

**Resources:** Tai Le, Xing Xia, Hung Cao.

**Software:** Daniel Jilani.

**Supervision:** Jimmy Zhang, Anh H. Nguyen, Ramses Seferino Trigo Torres.

**Validation:** Jimmy Zhang, Anh H. Nguyen.

**Visualization:** Jimmy Zhang, Anh H. Nguyen, Daniel Jilani, Ramses Seferino Trigo Torres.

**Writing – original draft:** Jimmy Zhang, Lauren Schmiess-Heine.

**Writing – review & editing:** Jimmy Zhang, Anh H. Nguyen, Hung Cao.

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
