## [Decision Letter · Decision Letter 0]

28 Feb 2023

PONE-D-23-03065Consecutive treatments of methamphetamine promote the development of cardiac pathological symptoms in zebrafishPLOS ONE

Dear Dr. Zhang,

Thank you for submitting your manuscript to PLOS ONE. After careful consideration, we feel that it has merit but does not fully meet PLOS ONE’s publication criteria as it currently stands. Therefore, we invite you to submit a revised version of the manuscript that addresses the points raised during the review process.

In general, both reviewers and myself found that there was a big jump between the findings of the manuscript and the conclusions made as a result of these findings. Therefore, either these 'jumps' need to be justified or there needs to be some toning down of the language used. Also, please ensure the spelling and grammatical errors picked up by the reviewers are also addressed

We look forward to receiving your revised manuscript.

Kind regards,

Daniel M. Johnson, PhD

Academic Editor

PLOS ONE

Journal Requirements:

4. Please amend the manuscript submission data (via Edit Submission) to include authors Dr. Lauren

Schmiess-Heine, Dr. Tai Le,  and Dr. Xing Xia.

Reviewers' comments:

Reviewer's Responses to Questions

**Comments to the Author**

1. Is the manuscript technically sound, and do the data support the conclusions?

Reviewer #1: No

Reviewer #2: No

2. Has the statistical analysis been performed appropriately and rigorously? 

Reviewer #1: N/A

Reviewer #2: No

3. Have the authors made all data underlying the findings in their manuscript fully available?

Reviewer #1: Yes

Reviewer #2: Yes

4. Is the manuscript presented in an intelligible fashion and written in standard English?

Reviewer #1: No

Reviewer #2: Yes

5. Review Comments to the Author

Reviewer #1: This report by Zhang and colleagues aims to elucidate the effect of methamphetamine (Meth) on cardiac physiology and electrophysiology in a standardized and controlled setting. To this end, the authors exposed adult zebrafish to meth and measured cardiac function using electrocardiography. This analysis suggests that treatment with Meth induces electrical alterations in the zebrafish heart, which the authors suggest are associated with changes in cAMP in cardiomyocytes.

While we acknowledge the authors’ hard work, several weaknesses in the experimental design and the interpretation of the results enormously reduce our enthusiasm. In short, the authors fail to justify their experimental setup, present minimal data and lack the required results to support most of their claims. For example, the authors claim that their goal is to “elucidate the mechanisms by which Meth induces electrophysiological abnormalities,” but they only explored changes in cAMP. Moreover, the results are poorly integrated with the text, which would benefit from extensive English revisions and editing. The following criticisms do not cover all the detected issues and should not be considered an exhaustive list. However, it might be a good starting point for the authors to revisit their results if they decide to resubmit their manuscript elsewhere.

MAJOR COMMENTS:

(1) Overstatements and unjustified claims. The main problem of this manuscript is the lack of data to support most of the claims. Indeed, the experimental data presented here is rather meager. Illustrative examples:

Line 331: “[we have sought to] Elucidate the mechanism of Meth in inducing electrophysiological abnormalities.” This study does not elucidate this. It characterizes some of the electrophysiological abnormalities observed after meth intake, but does not resolve how they have been generated. Showing an increase of cAMP signaling after meth treatment is not sufficient to make that claim. In this study, they show it is correlative, they don't demonstrate that it is causative.

Line 355-356: “To the best of our knowledge, this is the first time sinus block has been indicated as a symptom of Meth use.” This has never been demonstrated.

Line 425: “We propose that cAMP upregulation may be regulated by GPCRs such as TAAR1, as depicted in Fig 5A.” The data presented in this manuscript do not support this model though.

Line 432-433: “The results in this study also indicated that the upregulation of cAMP occurred in a dose-dependent manner and can be inhibited by an antagonist, displaying the plausible reversibility of Meth-induced cardiotoxicity”. This conclusion cannot be drawn from the presented results.

(2) Experimental Design and Interpretation. Another important problem is the apparently arbitrary experimental design, including the lack of appropriate controls. The authors are treating animals with Meth, and then measuring defects in electrical activity and cAMP, but the characterization is incredibly superficial. Specific examples:

• Is this set-up standard to do electrophysiological recording? Why the need to come up with their own?

• Please justify treatment concentrations and durations: please add justification on the dose of Meth used: why 200 µm? Why 20 min of treatment at 200 µm? Why 3 times a week? Does that correlate somehow with drug intake in humans? Why quantification only for two weeks?

• Figure 5: HEK293 used as control to test for cell specificity. I am not sure this is the most appropriate control. Perhaps a better one to claim that meth specifically induces cardiotoxicity would have been to isolate other cells types from treated fish and do the same quantification.

• Not clear how the dose-dependent experiments have been performed in Figure 5C-D: same cells subjected to different increasing doses, or different cells?

1) Rigor and Reproducibility of the Data.

• In figure 4: what would be the explanation for the drop of HR also in untreated fish at day 8 compared to other days (which looks significant to other days in untreated fish)?

• Figure 4B: For this quantification, it looks like the HRV is actually also increasing after 10 days in untreated fish. Please comment.

• Figure 4C: For this result, it is worth noting that based on the quantification, it actually looks like the PR stayed pretty constant in Meth-treated fish (~55ms), and that the differences between the two experimental groups seem to actually be carried by variations of the PR in the untreated ones along the course of the experiment (ranging from ~55 up to 69 ms). Explanations?

MINOR COMMENTS:

Figure 1-3 should be merged. They have only two panels each (A,B). Poorly integrated and discussed in the text. Some conclusions only described in figure legends.

In figure 5, forskolin is written with an e at the end, please correct.

Please specify somewhere in the manuscript what forskolin does.

Legend Figure 4: (C) Meth treatment induced a significant decrease during week 1 but not week 2 of treatment. Please specify what is decreasing in the legend.

Not clear why the potential effect of tricaine is being brought up in the discussion when the authors do not address that at all experimentally.

Line 32, in abstract: replace “previous” with “previously”.

Line 33, in abstract: typo: replace “has” with “have”, as assays plural.

Line 37, in abstract: replace “and remain”, with “that remained”.

Line 45, in abstract: add s to increases, as Methamphetamine is singular.

Line 91: Awkward phrasing: “it necessitates the research to”

Line 92-94: Quite the statement. I am sure a lot of non-zebrafish people would argue that it is not essential to understand meth abuse to fish to understand meth-induced cardiac toxicity.

Line 98: replace “its” by “their”

Line 103: “of” to be replaced by “to”?

Line 103: specify contractility of what; cell? Cardiomyocytes?

Line 110: might be worth specifying the remodeling of what.

Line 115: add “to” after “due”

Line 120: Similarity to what? to the study ref24? If not, please specify similarity to what. If when, it might add clarity to the sentence to replace "used for" to "to".

Line 124: They did not really look at GPCRs per se though.

Line 133: Might be worth specifying the genetic background of WT.

Line 143-144: Awkward redundancy, fix?

Line 179: Please add “be” after “can” to have “can be accessed”.

Line 234: I think “desiderate” should be replaced by “desiderated” in this sentence

Line 234: GloSensor instead of Glosensor for consistency

Line 249: What is the maximum effect expected in this experimental setting?

Line 258: Number of animals quite low for this study (CTL= 6; treated= 8).

Reviewer #2: In this manuscript the authors describe a preparation and a study in which they record cardiac activity from adult zebrafish following methamphetamine treatment. The goal of the work was to examine the relationship between cardiotoxicity and arrhythmia with associated methamphetamine exposure in adult zebrafish. Animals were treated with Meth 3 times a week for 2 weeks and multiple ECG measurements were taken. The authors also examined cAMP levels via a cAMP assay to ascertain the effect on this second messenger following Meth treatment.

I am reasonably convinced that meth acts either directly or indirectly on cardiac activity, but I am more skeptical about the mechanism and the nature of the interactions. The authors have not provided convincing evidence that zebrafish either have TAAR1 and that it is expressed in cardiac tissue (PCR or immunohistochemistry if antibodies exist). I understand that antagonists for TAAR1 are limited but a compound such as EPPTB which (I believe) is an inverse agonist might provide some information on the actions of Meth on TAAR1 in this study. I have also found a number of grammatical and spelling errors throughout the manuscript and respectfully suggest that the authors pay particular attention to these details.

Additional comments are listed below.

Comments:

1. Abstract, Line 31-32: the sentence that reads “zebrafish underwent electrocardiogram (ECG)…” needs to be adjusted because many readers may not know what the previously developed acquisition system is. In other words, clarify what the authors are referring to.

2. Abstract, line 36-38: Should probably read “…exhibited a decrease in heart rate at the end of the first week of treatment, which remained significantly lower than untreated fish in the second week.

3. Introduction, line 97-98: It may be how I read the sentence but to me this sentence implies that all GPCRs use cAMP as a second messenger. Please adjust for accuracy.

4. What is known about TAAR1 in zebrafish? can the authors provide evidence (maybe a reference or PCR etc) for TAAR1 expression in zebrafish?

5. Methods, line 155: I think this is the first time PDMS is used. Please explain what this is.

6. Methods, line 169: This sentence sounds a bit odd. Please review and correct if necessary.

7. Methods, line 219: Please explain what is BDM?

8. Methods line 241: Please explain what is the JMP suite?

9. Results: I found little indication of numbers of experiments and animals throughout the Results section or in the Figure legends.

10. Acknowledgements: It seems odd to me to Acknowledge persons who are already listed as Authors of the manuscript.

11. Figure 4: Apologies if I missed this but can the authors Discuss why there are some effects at intermediate time periods (eg Fig 4C Day 5) but not at earlier or later time periods (Day 3 or Day 8, 10).

12. Figure 5: Methamphetamine, Forskolin and Arrhythmia are misspelled. Please double check all spelling.

6. PLOS authors have the option to publish the peer review history of their article (what does this mean?). If published, this will include your full peer review and any attached files.

Reviewer #1: No

Reviewer #2: No

---

## [Author Response · Author response to Decision Letter 0]

20 Jul 2023

Dear Editors and Reviewers,

We appreciate your valuable comments and suggestions. We have carefully revised our manuscript following your comments point by point. The revisions are highlighted by mark-ups found within the file labeled ‘Revised Manuscript with Track Changes’. The clean copy of the revised manuscript is labeled as 'Manuscript'. Your constructive comments are extremely helpful for us to improve our work as well as to better revise the current manuscript.

Please find our detailed answers to reviewers’ comments in the file labeled 'Response to Reviewers'.

Best regards,

J. Zhang et al.

Department of Biomedical Engineering

University of California, Irvine

---

## [Decision Letter · Decision Letter 1]

31 Oct 2023

Consecutive treatments of methamphetamine promote the development of cardiac pathological symptoms in zebrafish

PONE-D-23-03065R1

Dear Dr. Cao,

We’re pleased to inform you that your manuscript has been judged scientifically suitable for publication and will be formally accepted for publication once it meets all outstanding technical requirements.

Kind regards,

Nicholas Aderinto Oluwaseyi

Academic Editor

PLOS ONE

Additional Editor Comments (optional):

Reviewers' comments:

Reviewer's Responses to Questions

**Comments to the Author**

1. If the authors have adequately addressed your comments raised in a previous round of review and you feel that this manuscript is now acceptable for publication, you may indicate that here to bypass the “Comments to the Author” section, enter your conflict of interest statement in the “Confidential to Editor” section, and submit your "Accept" recommendation.

Reviewer #3: All comments have been addressed

2. Is the manuscript technically sound, and do the data support the conclusions?

Reviewer #3: Yes

3. Has the statistical analysis been performed appropriately and rigorously? 

Reviewer #3: Yes

4. Have the authors made all data underlying the findings in their manuscript fully available?

Reviewer #3: Yes

5. Is the manuscript presented in an intelligible fashion and written in standard English?

Reviewer #3: Yes

6. Review Comments to the Author

Reviewer #3: (No Response)

7. PLOS authors have the option to publish the peer review history of their article (what does this mean?). If published, this will include your full peer review and any attached files.

Reviewer #3: No

---

## [Editor Report · Acceptance letter]

9 Nov 2023

PONE-D-23-03065R1 

Consecutive treatments of methamphetamine promote the development of cardiac pathological symptoms in zebrafish 

Dear Dr. Cao:

I'm pleased to inform you that your manuscript has been deemed suitable for publication in PLOS ONE. Congratulations! Your manuscript is now with our production department. 

Kind regards, 

on behalf of

Dr. Nicholas Aderinto Oluwaseyi 

Academic Editor

PLOS ONE